# Physiological Response to Thermal Stress in Obese vs. Non-Obese Physically Inactive Men

**DOI:** 10.3390/biology11030471

**Published:** 2022-03-18

**Authors:** Robert Podstawski, Krzysztof Borysławski, Andrzej Pomianowski, Imre Soós, Michał Boraczyński, Piotr Gronek

**Affiliations:** 1Department of Tourism Recreation and Ecology, University of Warmia and Mazury in Olsztyn, 10-719 Olsztyn, Poland; 2Institute of Health, Angelus Silesius State University, 58-300 Wałbrzych, Poland; kboryslawski@puas.pl; 3Department of Internal Diseases with Clinic, University of Warmia and Mazury in Olsztyn, 10-719 Olsztyn, Poland; apomian@uwm.edu.pl; 4Faculty of Health Sciences, Doctoral School of Health Sciences, University of Pécs, H-7622 Pécs, Hungary; imre.soos@eto.hu; 5Department of Health Sciences, Collegium Medicum, University of Warmia and Mazury, 10-561 Olsztyn, Poland; michal.boraczynski@gmail.com; 6Laboratory of Healthy Aging, Department of Dance, Poznań University of Physical Education, 61-555 Poznań, Poland; gronek@awf.poznan.pl

**Keywords:** Finnish sauna, thermal stress, sedentary obese men, physiological parameters, body composition

## Abstract

**Simple Summary:**

It is important to determine the effects of thermal stress on the physiological parameters of obese persons. The aim of this study was to analyze the influence of sauna bathing on obese and non-obese physically inactive men. Sixty volunteers aged 18–24 years were divided into two groups (group I—normal body mass, group II—class 1 obesity). Their physiological parameters were monitored before and during the sauna session. The average values of body mass, body mass index, body surface area, and weight-to-height ratio were significantly higher in obese men than in the normal weight group. The values of physiological parameters were also significantly higher in obese individuals. However, the observed changes remained within the norm, which indicates that a 10 min sauna session is safe for young men regardless of their body fat levels.

**Abstract:**

The effects of thermal stress on the physiological parameters of obese subjects remain insufficiently researched. The objective of this study was to determine the influence of sauna bathing on the physiological parameters of obese and non-obese physically inactive men. Sixty volunteers aged 18–24 years (20.85 ± 1.46) were divided into two groups (group I—normal body mass, group II—class 1 obesity) for a comparative analysis. Somatic features and body composition were determined before sauna, and blood pressure was measured before and after sauna. Physiological parameters were monitored before and during the sauna session. The average values of body mass, body mass index, body surface area, and weight-to-height ratio were significantly higher (*p* < 0.001) in obese men (by 28.39 kg, 8.7 kg/m^2^, 0.34 m^2^, and 0.13, respectively) than in the normal weight group. Similar observations were made in an analysis of the remaining body composition parameters. The values of physiological parameters (heart rate, systolic and diastolic blood pressure, energy expenditure, oxygen uptake, excess post-exercise oxygen consumption, respiratory rate) were significantly (*p* < 0.001) higher in obese subjects. However, the observed physiological changes were within the expected norm; therefore, a 10 min sauna session is safe for young men regardless of their body fat levels.

## 1. Introduction

Growing levels of awareness about health and the benefits of a healthy lifestyle have contributed to the popularity of sauna bathing in recent years [1]. Sauna is an ancient traditional form of whole-body heat therapy that relies on contrast thermal effects. Sauna bathing takes place at a temperature of 70–100 °C (the optimal range is 80–90 °C) and low relative humidity of 15–30% [2]. Bathers are exposed to repeated cycles of heat and cold. After exposure to heat, bathers take a cold shower or swim in a cold paddling pool to decrease their body temperature [3].

There is a general consensus that regular sauna visits improve vascular and cardiac functions [4,5,6]. Sauna bathing significantly influences the function of many bodily organs and systems [7,8]. In healthy subjects, sauna improves cardiovascular function, regulates hemodynamic processes, promotes coronary circulation, and stabilizes arterial blood pressure [9]. When used appropriately, sauna has a positive effect on skeletal muscles by speeding up the excretion of metabolic waste [10]. A long-term prospective study revealed that the frequency and duration of Finnish sauna bathing play a role in preventing cardiovascular and all-cause mortality, sudden cardiac death (SCD), dementia, and Alzheimer’s disease [11,12].

The main physiological response to sauna bathing is an accelerated heart rate (HR), which increases to twice the resting HR or higher [13]. According to estimates, cardiac output increases by approximately 70% relative to rest, whereas total peripheral vascular resistance decreases by approximately 40% [8]. Diastolic and mean arterial pressure decrease, but changes in systolic pressure are minimal or non-existent [14].

The effects of thermal stress on the physiological parameters of obese subjects have been far less widely researched. The number of studies investigating the effects of thermal stress is further narrowed down when variables such as sex, physical activity (PA) levels, and absence of regular sauna use are taken into account. Rapid physiological changes can be expected in persons who have never visited a sauna [15]. Obese individuals are also likely to respond differently to thermal stress than persons with normal body mass. The aim of this study was to identify the physiological changes induced by a 10 min session in a Finnish sauna in young sedentary men with class 1 obesity who have never visited a sauna and to compare the results with non-obese physically inactive peers.

## 2. Materials and Methods

### 2.1. Participant Selection

The sampling process was carried out in stages, and potential candidates had to meet specific criteria. Their PA levels were assessed using the International Physical Activity Questionnaire (IPAQ) based on the average number of minutes dedicated to PA per week (minimum 10 min). The energy expenditure associated with the reported activities was expressed in metabolic equivalent of task (MET) units. The MET is the ratio of working metabolic rate relative to resting metabolic rate, and 1 MET denotes the amount of oxygen utilized by a person per minute, which is estimated at 3.5 mL/kg/min [16]. Based on the METs minutes, the participants were divided into groups representing low (L < 600 METs-min/week), moderate (M < 1500 METs-min/week), and high (H ≥ 1500 METs-min/week) levels of activity. Only male students characterized by low levels of PA were selected for the study. Women were not included in the study because their number was insufficient. Only several women who had met the relevant inclusion criteria volunteered to participate in this trial. To be included in this study, the participants could not take any medications or nutritional supplements before the study and could not have a history of respiratory or circulatory diseases. A total of 123 volunteers were willing to participate in the study, and those who met the inclusion criteria were notified of the date of final recruitment by e-mail. Ultimately, the study was conducted on 60 male volunteers who were full-time university students aged 18–24 years (20.85 ± 1.46) and had never visited a sauna before the study.

### 2.2. Ethical Approval

Prior to the study, an ethical approval was obtained from the Ethics Committee of the University of Warmia and Mazury in Olsztyn (UWM), Poland (No. 10/2020). The participants were student volunteers who signed an informed consent statement.

### 2.3. Instruments and Procedures

The participants received comprehensive information about sauna rules during PE classes preceding the study. They were asked to eat a light meal no later than 3 h before sauna and drink at least 1 L of water on the day of the test, including 0.5 L of water 2 h before the session. Dry sauna sessions were held weekly during PE classes, on the same day, in the same location, and over the same period of time to minimize the effect of diurnal variations on the results [17]. Every participant attended one 10 min sauna session (temperature: 90 °C; relative humidity: 14–16%) during which they remained in a sitting position. After the 10 min session, the subjects rested for 10 min in a room with a temperature of 18 °C. The volunteers took a shower set to a temperature of 14–15 °C during the recovery session. They could also cool down in a paddling pool (water temperature: +10 °C).

Before the first sauna session, body height was measured to the nearest 0.1 cm with a stadiometer. Anthropometric measures, including body mass (measured to the nearest 0.1 kg), waist, and hip circumferences were taken using a calibrated scale and tape measure. From these measurements, body mass index (BMI), body surface area (BSA), and the waist–hip ratio (WHR) were calculated. Body composition parameters, including total body water (TBW), protein and mineral content, body fat mass (BFM), fat-free mass (FFM), skeletal muscle mass (SMM), percent body fat (PBF), InBody score, target weight, visceral fat level (VFL), and the degree of obesity were determined by bioelectrical impedance with the InBody 720 body composition analyzer [18]. During exposure to high temperature in the sauna, physiological parameters, including heart rate (HR _min_, _avg_, _max_), recovery time, peak training effect (PTE), energy expenditure, estimated oxygen uptake (VO_2_
_avg_, _max_), estimated excess post-exercise oxygen consumption (EPOC _avg_, _peak_), estimated respiratory rate (RR _avg_, _max_), and physical effort (easy, moderate, difficult, very difficult, maximal) were measured with Suunto Ambit3 Peak Sapphire (Vantaa, Finland) heart rate monitors, which are widely used in studies of the type [19]. Heart rate monitors were placed on the wrist, and HR monitor sensors were attached to the chest. Every pulsometer was programed to male sex, year of birth, body mass, and PA level. Every participant activated the heart rate monitor for 10 min before entering the sauna. Upon exiting the sauna, the participants deactivated their heart rate monitors and switched to the data save mode. Minimal, average, and maximal heart rate, along with physical efforts associated with these values, were recorded. Blood pressure (BP) measurements were performed with the participant sitting on a medical chair before and immediately after the sauna session. A digital blood pressure monitor (Omron M6 Comfort, Kyoto, Japan) was used to measure BP.

### 2.4. Statistical Analysis

Anthropometric, body composition, and physiological parameters were processed using the Statistica PL v. 13.5 application with the use of descriptive statistics. The participants were divided into two groups of 30 subjects each for a comparative analysis. Group I (control) consisted of males with normal body fat levels, and group II was composed of males with class 1 obesity. The arithmetic means of all parameters measured in both groups were compared in the Student’s *t*-test for independent groups. All analyzed parameters had normal distribution. Normality was verified with the Shapiro–Wilk test. The asymmetry coefficient (As) was also calculated. Significance was set at the 95th level of confidence (*p* ≤ 0.05).

## 3. Results

The descriptive statistics for anthropometric parameters and body composition parameters are presented in Table 1. Significant deviations from normal distribution were not observed in any of the investigated parameters.

The evaluated subjects did not differ significantly in average age or average body height, which indicates that the studied sample was homogeneous. Significant differences in body mass and the associated parameters (BMI, BSA, and VHR) were observed between groups. The average body mass, BMI, BSA, and VHR were significantly higher (*p* < 0.001) in obese subjects (28.39 kg, 8.7 kg/m^2^, 0.34 m^2^, and 0.13 on average, respectively). The average BMI values in group II (30.92 kg/m^2^) were indicative of class 1 obesity, and the maximum BMI (40.2 kg/m^2^) was indicative of class 3 obesity. The average WHR (0.94) in group II exceed the reference value by 0.4 and was indicative of gynoid obesity (fat accumulation in hips and thighs), but the reference value for android obesity (abdominal fat) was not exceeded. The values of TBW, proteins, minerals, SMM, PBF, BFM, FFM, and VFL were significantly higher (*p* < 0.001) in group II (by 7.28 L, 2.04 kg, 0.71 kg, 6.2 kg, 13.24 kg, and 1.5 kg on average, respectively) than in group I. The values of BFM control differed significantly between groups (*p* < 0.001) and indicated that group I subjects should lose 0.58 kg of body fat, whereas group II subjects should lose 16.57 kg of body fat to achieve healthy BFM levels. The values of FFM control (despite the presence of significant differences) indicated that group I subjects could benefit from increasing their FFM by 1.57 kg, whereas no recommendations were made in group II. The average body mass was 0.99 kg below the suggested target weight in group I, and it was significantly higher (*p* < 0.001) at 16.57 kg above the suggested target weight in group II (Table 2).

All analyzed HR values (min, avg, max) were significantly higher (0.001 < *p* < 0.013) in obese subjects. The values of HR max were very similar in both groups (129–130 bpm) and reached the range indicative of very difficult physical effort (125–141 bpm). Average energy expenditure was 66.80 kcal in group I, and it was significantly higher (*p* < 0.001) at 92.37 kcal in group II. Similar trends were observed in the values of VO 2(avg, max), EPOC (avg, Peak), respiratory rates (avg, max), SBP (before sauna, after sauna, increase), and DBP (after sauna, increase). A minor increase in the average values of SBP and DBP (by 0.87 and 0.77 mmHG, respectively) was noted in group I, whereas a greater increase (by 2.40 and 2.13 mmHG, respectively) was observed in group II. Interestingly, in some cases, SBP and DBP decreased below the initial values determined before sauna. Despite elevated BP levels (SBP and DBP) among obese subjects, SBP and DBP were within the norm (<140/90 mmHG). During a 10 min sauna session, HR values were within the range indicative of easy to difficult physical effort. Group I males remained within the easy effort range significantly (*p* < 0.001) longer (507.07 s) than group II males (384.27 s), whereas obese subjects remained within the moderate effort range significantly longer (211.10 s) than males with normal body mass (91.37 s).

## 4. Discussion

The present study delivered highly interesting results. Our findings were interpreted based on the relevant literature to expand the existing knowledge about the ways in which thermal stress influences the human body when obesity is taken into account in subjects matched for sex, physical activity, and experience in using a dry sauna.

As expected, the comparative analysis of the basic anthropometric parameters showed that body composition parameters were significantly different between normal weight and overweight subjects. The physiological parameters revealed significant differences in the HR values of group II subjects, clearly indicating that obese males using a dry sauna are much more susceptible to high temperature than subjects with normal body mass. Other studies have reported the HR of young people who regularly visit the sauna increasing to 100–110 bpm, although it can exceed 140–150 bpm with a rise in sauna temperature [20,21,22]. In this study, the subjects were first-time sauna users, which is why significant differences were captured even during a short sauna session of only 10 min. A similar relationship was reported by Leppaluoto et al. [23] who found a greater increase in the HR of sporadic sauna users and attributed this observation to the lack of physiological adaptation to high temperature. In our study, the tested subjects’ physiological adaptation could have also been compromised by low PA levels, which might have an effect on the response to thermal stress. Other factors, such as the duration of the sauna session, age, sex, and physical endurance also influence HR values.

In terms of health outcomes, the adaptive response is regarded as beneficial when the HR increases to around 120 bpm, but an increase in excess of 140 bpm leads to higher cardiac effort and diastole shortening, which could have adverse consequences for health [3]. In this study, the maximum HR values did not exceed 130 bpm, which suggests that a 10 min sauna session (temperature: 90 °C; relative humidity: 15%) is safe for subjects with class 1 obesity.

The average values of HR_min_ measured before sauna were significantly higher in group II (85.63 bpm) than in the control group (78.60 bpm), which could be attributed to psychological discomfort associated with the first sauna visit. Sauna bathing at a temperature of 90 °C can lead to a massive physical effort, and not all users experience physiological and psychological benefits. In our previous study [24], most respondents claimed that sauna bathing delivered relaxing and calming effects, but in another study [25], 23.7% men and 14.93% women experienced discomfort caused by high temperature, claustrophobia, a high number of sauna users, and the presence of the opposite sex in the sauna room. Our previous study [26] revealed that the effect of sexual dimorphism was linked with the sociocultural status of participants. Similar values of HR (82.7 bpm) before sauna use were also found in men who visited a sauna only sporadically and who were not professional athletes. Significantly lower HR values were observed in male subjects with average and high training levels (71.8 bpm and 68 bpm, respectively) who regularly visited a sauna [27]. Pilch et al. [15] reported that the HR values of 10 professional swimmers and 10 untrained students (aged 20–23 years) increased from 74 and 108 bpm before sauna use to 133 and 144 bpm, respectively, after three 15 min sauna sessions with 5 min breaks (temperature: 92.3 °C, humidity: 27.4%). In a follow-up study, the HR values of 10 males (aged 25–28 years) increased significantly from 66.6 bpm before sauna use to 126 bpm after three 15 min dry sauna sessions with 5 min breaks [28], which confirms that the duration of exposure to thermal stress plays a significant role. In males with class 1 obesity, an increase in HR leads to higher energy expenditure and higher values of physiological parameters, such as VO_2_
_avg_, _max_, EPOC _avg_, _peak_, and respiratory rate _avg_, _max_. Previous research evaluating the effect of thermal stress on the physiological parameters of young, overweight, and sedentary men who visited a sauna only occasionally revealed that increasing values of correlation coefficients during successive sauna sessions point to stronger correlations between the examined anthropometric characteristics and indicators (such as body mass, BMI, BSA, WHR) and physiological parameters (HR _min_, _mean_, _max_, energy expenditure, VO_2_
_avg_, _max_, EPOC _avg_, _peak_, respiratory rate _avg_, _max_, and BP _SBP_, _DBP_) during prolonged sauna use [29]. Similar relationships were also observed in a study exploring the effect of prolonged thermal stress on the physiological parameters of young, overweight, and sedentary men [30].

Blood pressure (SBP and DBP) is also an important physiological parameter. The results of studies investigating the influence of sauna bathing on BP differ considerably subject to measurement method, type of sauna, duration of exposure, which elicits the evaporation effect, and user adaptation to high temperature [3]. In the literature, the results of BP measurements performed with a sphygmomanometer differed considerably and ranged from a minor increase [23] or the absence of any changes [31], to a decrease in SBP [6,32] and DBP values [33]. In the current study, a minor increase in SBP and DBP was noted in both groups. However, BP measured before and immediately after sauna was significantly higher in obese subjects, and the increase in BP values was also significantly higher in group II than in the control group. Obese males were generally characterized by higher SBP and DBP values than subjects with normal body mass [34]. High SBP and DBP values can be indicative of hypertension and other co-morbidities [35]. According to medical guidelines, hypertensive individuals should use the sauna at lower temperatures in the range of 45–50 °C. This temperature range is encountered in a steam sauna, but exposure to high humidity (100%) is not recommended either [3]. Significant changes in BP values were reported during sauna sessions lasting 30 min and longer. The SBP values of men with various training levels (high, average, and men who did not train professionally) increased (from 125.7 to 133, from 121.5 to 129.6, from 113.4 to 119.4 mmHg), whereas their DBP values decreased during sauna (from 73.47 to 69.4, from 75.6 to 73.5, from 71.4 to 70.1 mmHg, respectively) [27]. In a study of 10 healthy males aged 25–28 years [28], SBP values increased from 122.6 to 142.6 mmHg, and DPB values decreased from 78.7 to 63.7 mmHg after three 15 min sauna sessions separated by 5 min breaks (temperature: 92.3 °C, humidity: up to 17.4%). A significant decrease in SBP values (112 ± 10 mmHg, *p* = 0.013) without any changes in DBP values was observed after a 30 min session in a dry sauna (65 °C) [36]. In men aged 22–53 years who bathed in a sauna for 30 min at a temperature of 80–90 °C, SBP values increased from 118 to 120 mmHg [37]. The results of this study and the findings reported by other authors who investigated the effects of sauna sessions lasting 30 min or longer indicate that 10 min sauna sessions do not pose health risks for young and obese males who are first-time sauna users.

### Limitations

The physiological parameters of the participants were measured with Suunto Ambit3 Peak Sapphire heart rate monitors, which could be a potential limitation of this study. However, the participants were exposed to extreme temperature (90 °C), and other types of measuring equipment could not be used to analyze a large and homogenous sample (2 groups of 30 males each) under similar environmental conditions (day, hour, duration, temperature, and humidity) in an effective and safe manner. The reliability and validity of different HR monitors under extreme conditions may be examined in the future. Further research involving a larger sample of physically active men is also needed to confirm the present findings.

## 5. Conclusions

Exposure to extreme thermal stress during a 10 min sauna session (temperature: 90 °C, humidity: 15%) induced significantly more pronounced physiological changes in young males with class 1 obesity than in subjects with normal body mass. The values of physiological parameters (energy expenditure, HR _min_, _avg_, _max_, SBP, DBP, VO_2_
_avg_, _max_, EPOC _avg_, _peak_, and respiratory rate – RR _avg_, _max_) were significantly higher in young, sedentary, and obese men than in their peers with normal body mass. However, the physiological parameters measured before and after sauna were within the norm in both groups, which suggests that exposure to thermal stress during a 10 min sauna session (temperature: 90 °C, humidity: 15%) is safe even for men with class 1 obesity. The findings from the current study complement previous research and contribute to a better understanding of the factors that significantly affect physiological parameters during sauna bathing. Our findings should be considered by individuals, practitioners, and clinicians to ensure safe sauna exposure with maximum health benefits.

## Figures and Tables

**Table 1 biology-11-00471-t001:** Descriptive statistics of anthropometric parameters and body composition parameters in males with normal (relative) body mass (*n* = 30) and in obese subjects (*n* = 30).

Parameters	BMI-Normal (*n* = 30)				BMI-Obese (*n* = 30)				*t*-Test
	Mean	SD	Min–Max	As	Mean	SD	Min–Max	As	
Age (years)	20.50	1.33	19–23	0.65	21.20	1.52	18–24	−0.17	-
Body height (cm)	180.63	6.12	170–192	−0.12	180.67	8.93	166–196	0.04	-
Body mass (kg)	72.50	6.83	55.9–84.0	−0.50	100.89	16.93	76.9–137.7	1.11	**−8.52**
Weight control (kg)	0.99	4.12	−5.4–8.4	0.39	−16.57	11.00	−50.8–−4.8	−2.15	**8.19**
BMI (Body Mass Index) (kg/m^2^)	22.22	1.72	19.2–24.8	−0.28	30.92	3.38	27.9–40.2	1.53	**−12.56**
BSA (Body Surface Area) (m^2^)	1.91	0.11	1.63–2.11	−0.64	2.25	0.23	1.89–2.67	0.76	**−7.24**
WHR (Waist–Hip Ratio)	0.81	0.03	0.72–0.84	−0.91	0.94	0.08	0.87–1.20	1.26	**−10.78**
TBW (Total Body Water) (L)	45.14	5.05	32.9–56.1	−0.46	52.42	6.46	39.4–64.0	0.06	**−4.87**
Proteins (kg)	12.22	1.38	8.9–15.0	−0.51	14.26	1.71	10.6–17.2	−0.02	**−5.09**
Minerals (kg)	4.26	0.54	3.24–5.31	−0.10	4.97	0.73	3.55–6.25	0.11	**−4.23**
SMM (Skeletal Muscle Mass) (kg)	34.84	4.16	24.7–43.0	−0.54	41.04	5.18	30.1–50.0	−0.01	**−5.12**
PBF (Percent Body Fat) (%)	15.07	4.33	6.0–23.3	0.00	28.31	6.24	19.3–46.3	1.42	**−9.56**
BFM (Body Fat Mass) (kg)	10.88	3.16	4.0–16.3	0.11	29.23	11.48	18.1–63.8	2.06	**−8.44**
BFM control (kg)	−0.58	2.92	−5.6–5.0	−0.08	−16.57	11.00	−50.8–−4.8	−2.15	**7.67**
FFM (Fat Free Mass) (kg)	61.63	6.95	45.0–76.4	−0.45	71.67	8.89	53.6–87.5	0.05	**−4.87**
FFM control (kg)	1.57	2.53	0.0–9.1	1.61	0.00	0.00	0.0–0.0	0.00	**3.40**
VFL (Visceral Fat Level) (kg)	3.80	1.54	1–7	0.23	12.13	5.30	7–28	2.08	**−8.28**
InBody score	78.40	5.91	69–89	0.34	74.27	11.81	39–91	−1.64	-
Target weight	73.49	5.88	63.6–84.0	−0.23	84.32	10.46	63.0–103.0	0.05	**−4.94**
PA levels in METS (3.5 mL/kg/min)	512.13	72.76	390–596	−0.56	465.40	58.53	390–580	0.59	2.74

Note: only statistically significant values of differences (Student’s *t*-test) are given; normal font denotes *p* values 0.05–0.001 and bold font *p* values ≤ 0.001.

**Table 2 biology-11-00471-t002:** Descriptive statistics of physiological parameters in men with normal (relative) body weight (*n* = 30) and in obese subjects (*n* = 30).

Parameters		BMI-Normal (*n* = 30)				BMI-Obese (*n* = 30)				*t*-Test
		Mean	SD	Min–Max	As	Mean	SD	Min–Max	As	
Heart rate-HR (bpm)	Min	78.60	6.70	67–96	0.91	85.63	8.21	58–97	−1.25	**−3.63**
	Avg	94.73	6.57	84–107	0.27	102.60	3.60	95–111	0.22	**−5.75**
	Max	113.50	9.95	92–130	−0.27	118.87	5.67	106–129	−0.52	−2.57
	Increase	34.90	11.36	17–56	0.07	33.23	10.13	20–58	0.44	-
Recovery time (h)		0.17	0.38	0–1	1.88	0.43	0.50	0–1	0.28	−2.32
Peak training effect-PTE		1.19	0.11	1.0–1.4	0.14	1.30	0.11	1.2–1.6	1.05	**−3.89**
Energy expenditure (kcal)		66.80	9.11	52–87	0.15	92.37	8.81	68–111	−0.31	**−11.05**
VO_2_ (mL/kg/min)	Avg	13.67	2.26	10–18	0.27	15.73	1.82	12–20	0.06	**−3.90**
	Max	20.03	3.71	13–26	−0.21	21.80	2.93	15–26	−1.09	**−2.05**
EPOC (mL/kg)	Avg	1.47	057	1–3	0.73	2.40	0.77	1–4	0.60	**−5.33**
	Max	3.53	1.48	2–7	0.55	5.23	1.37	3–9	0.89	**−4.60**
Respiratory rate (bpm)	Avg	16.87	1.31	15–19	−0.14	17.70	1.15	16–20	0.64	−2.62
	Max	22.60	1.33	20–25	−0.04	23.00	1.76	16–26	−2.02	-
Systolic blood pressure-SBP (mmHg)	Before sauna	122.23	7.92	106–134	−0.54	129.17	5.60	117–137	−0.57	**−3.92**
	After sauna	123.10	7.32	107–135	−0.52	131.57	5.96	118–140	−0.62	**−4.91**
	Increase	0.87	1.28	−3–3	−0.58	2.40	2.28	−1–12	2.54	−3.21
Diastolic blood pressure-DBP (mmHg)	Before sauna	79.53	5.11	69–89	−0.27	81.20	5.29	68–92	−0.43	-
	After sauna	80.30	5.23	70–94	0.14	83.33	5.49	75–96	0.12	−2.19
	Increase	0.77	2.03	−4–5	0.10	2.13	2.26	−4–7	−0.27	−2.47
**Physical effort (s)**										
Easy < 107 (bpm)		507.07	118.58	201–600	−1.30	384.27	106.20	152–594	−0.08	**4.23**
Moderate 107–124 (bpm)		91.37	118.31	0–399	1.35	211.10	103.55	6–447	0.11	**−4.17**
Difficult 125–141 (bpm)		1.57	5.30	0–25	3.80	4.63	17.11	0–92	4.93	-
Very Difficult 142–159 (bpm)		**All values are zero**								
Maximal ≥ 160 (bpm)		**All values are zero**								

Note: notations of significance of differences as in Table 1; normal font denotes *p* values 0.05–0.001 and bold font *p* values ≤ 0.001.

## Data Availability

The data presented in this study are available on request from the corresponding author. The data are not publicly available due to privacy.

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
