# Peer review of "Physiological Response to Thermal Stress in Obese vs. Non-Obese Physically Inactive Men"

_biology, 2022, doi:10.3390/biology11030471_

Round 1

Reviewer 1 Report

The presented manuscript "Physiological response to thermal stress in obese vs. non-obese physically inactive men” is largely well written and generally informative. There seems, however, to be a few minor concerns in this manuscript. No major comments as to the method of analysis and statistical research. The introduction and discussion are clearly written. However, the authors did not explain why only the group of men was qualified for the study. In the chapter Limitations, the authors should emphasize that the conclusions drawn should be verified by later studies on a larger group of people given thermal stress (obese vs. non-obese physically inactive men).

Author Response

Dear Reviewer,

Our response to reviews is attached in a file.

Sincerely,

Robert Podstawski

Reviewer 2 Report

The undertaken research topic is interesting and useful and may attract the attention of readers. The aim of the study is well presented. The manuscript is well organized. The results are clearly presented and discussed in the light of previous scientific reports. The conclusions are supported by the results. The English is sufficiently understandable.

A minor remark: page 7, line 261 is “temperature: 9.23°C” – it should probably be 92.3°C.

Author Response

(The authors gave the same response as above.)

Reviewer 3 Report

Dear Authors 
Thank you for the opportunity to review this manuscript. This paper is very interestingly and can be useful to scientific evidence updating. However, few aspect should be revised:
-    the introduction should be revised and updated with several detailed studies on epidemiological data on the effects of thermal stress on human health.
-    In material and methods at lines 131-134 the authors describe the groups used for statistical analysis, however in table 1 and 2 the groups are described in different way. The author should check the table and adequate change.

Author Response

Dear Reviewer,

Our response to review is attached in the file.

Sincerely,

Robert Podstawski
